# A Two-Mediator System Based on a Nanocomposite of Redox-Active Polymer Poly(thionine) and SWCNT as an Effective Electron Carrier for Eukaryotic Microorganisms in Biosensor Analyzers

**DOI:** 10.3390/polym15163335

**Published:** 2023-08-08

**Authors:** Anastasia S. Medvedeva, Elena I. Dyakova, Lyubov S. Kuznetsova, Vladislav G. Mironov, George K. Gurkin, Tatiana V. Rogova, Anna S. Kharkova, Pavel V. Melnikov, Alina O. Naumova, Denis N. Butusov, Vyacheslav A. Arlyapov

**Affiliations:** 1Research Center “BioChemTech”, Tula State University, 92 Lenin Avenue, 300012 Tula, Russia; 2M. V. Lomonosov Institute of Fine Chemical Technologies, MIREA—Russian Technological University, 119571 Moscow, Russia; 3Computer-Aided Design Department, Saint Petersburg Electrotechnical University “LETI”, 197022 Saint Petersburg, Russia

**Keywords:** redox-active polymers, poly(thionine), single-walled carbon nanotubes (SWCNT), nanocomposite, yeast *Blastobotrys adeninivorans*, biochemical oxygen demand (BOD)

## Abstract

Electropolymerized thionine was used as a redox-active polymer to create a two-mediated microbial biosensor for determining biochemical oxygen demand (BOD). The electrochemical characteristics of the conducting system were studied by cyclic voltammetry and electrochemical impedance spectroscopy. It has been shown that the most promising in terms of the rate of interaction with the yeast *B. adeninivorans* is the system based on poly(thionine), single-walled carbon nanotubes (SWCNT), and neutral red (k_int_ = 0.071 dm^3^/(g·s)). The biosensor based on this system is characterized by high sensitivity (the lower limit of determined BOD concentrations is 0.4 mgO_2_/dm^3^). Sample analysis by means of the developed analytical system showed that the results of the standard dilution method and those using the biosensor differed insignificantly. Thus, for the first time, the fundamental possibility of effectively using nanocomposite materials based on SWCNT and the redox-active polymer poly(thionine) as one of the components of two-mediator systems for electron transfer from yeast microorganisms to the electrode has been shown. It opens up prospects for creating stable and highly sensitive electrochemical systems based on eukaryotes.

## 1. Introduction

At present, biosensor technologies are quite often used in environmental monitoring since they allow for fast, accurate analysis of the state of the environment [1,2]. Often, standard methods are not capable of performing a rapid assessment, as they may require a long incubation. For example, the duration of the analysis is 5 days when assessing biochemical oxygen demand (BOD) [3], so there is a high demand for the development of biosensors for rapid assays [4,5,6]. The BOD index reflects the total content of organic pollutants in wastewater (mg/dm^3^) and shows the potential ability of wastewater to consume oxygen reserves in natural water. This parameter does not take into account what organic components the sample is contaminated with, but it allows one to quickly respond to pollution of natural waters as well as adjust the degree of wastewater treatment. Microorganisms with a wide range of oxidizable substrates are used for the formation of bioreceptors, taking into account the nature of the BOD index. The main problem in working with microorganisms is the high value of the lower limit of the range of determined BOD values, which exceeds the value of BOD in natural waters [4,7]. Various approaches are used to modify the electrodes to lower the values of this property, which characterizes the sensitivity of the system. One of these approaches is the creation of nanocomposite, which includes conductive polymers and nanomaterials [8,9,10,11,12].

Redox-active or conductive polymers are often applied for biological material immobilization. It is possible to create reagent-less biosensors using such biocompatible and simple materials. Comparing biosensors based on classical mediator systems, biosensors based on redox-active polymers usually outperform by operation stability and electron transfer rate; in a biosensor assay, the charged particles are not washing out from the biocatalytic layer [11,12,13]. It should be noted that biosensors based on redox-active polymers are characterized by weak attachment of the polymer to the working electrode surface. This causes a decline in electron transport efficiency. Nanomaterials are used to increase the surface area, improve the polymer conductivity, and improve its adhesion to the electrode [14]. Applying to biosensor analysis, carbon nanotubes (CNT) and other nanomaterials increase the electrocatalytic process rate [15,16,17,18].

Currently, there are a number of published works describing BOD biosensors based on redox-active or conductive polymers and their nanocomposites. For example, a mediator-type BOD biosensor was developed using polypyrrole (PPy), immobilized ferricyanide (FC) as a mediator, and *Pseudomonas aeruginosa* bacteria [19]. The FC mediator and *P. aeruginosa* microorganisms were introduced into the PPy matrix on the surface of a gold microelectrode during the electropolymerization of the pyrrole monomer using the cyclic voltammetry (CV) method. A similar approach was also used in another work on the creation of a BOD biosensor based on *Pseudomonas aeruginosa* bacteria [20]. Poly (neutral red) (pNR) was used as the redox-active polymer. The biosensor made it possible to estimate BOD in the range of 5–100 mg/dm^3^. In the study [21], *Bacillus subtilis* bacteria were immobilized in a three-dimensional porous graphene-polypyrrole composite material, and potassium hexacyanoferrate (III) was used as a mediator. The range of measurable BOD_5_ values was 4–60 mgO_2_/dm^3^, and the system was stable for 60 days. It should be noted that currently developed microbial biosensors based on similar conductive materials exhibit poor performance, although this approach is very effective in the field of enzyme biosensors [7,10,22]. In addition, these examples refer to the use of bacterial microorganisms. Yeast is much more efficient for BOD analysis due to a wider range of oxidizable substances [4,23]. However, their conjugation with conductive polymers in biosensor devices is a very difficult task due to their intracellular localization of enzyme systems and a thick cell wall, and at the moment such works are not presented in the literature. Thus, the search for optimal solutions for the formation of sensitive, selective, and cost-effective microbial biosensors for determining BOD continues to this day. In addition, BOD analysis is a classic model experiment in which new approaches to the formation of receptor systems based on microorganisms could be compared. In our current work, we propose and test for the first time a new approach to the creation of such systems by forming two mediator systems, one of whose components is a conducting nanocomposite material based on electrochemically polymerized poly(thionine) and CNT.

## 2. Materials and Methods

### 2.1. Reagents and Equipment

Glucose, tryptone, peptone (Panreac, Barcelona, Spain), yeast extract (Helicon, Moscow, Russia), and agar-agar (Panreac, Spain) were used to grow microorganisms. Graphite powder with a particle size of 75 μm and a purity of 99.997% (Fluka, Neu-Ulm, Germany), paraffin oil (Fluka, Germany), and a dialysis membrane with a 14 kDa transmission limit (Roth, Germany) are used for forming the working electrode. Thionine (Dia-m, Moscow, Russia) was used as the basis of the redox-active matrix. Single-walled CNT (SWCNTs) (Uglerod Chg, Moscow, Russia, CNT length 1–10 µm, average diameter 1.5 nm, external specific surface 450 m^2^/g), multi-walled CNT (MWCNT) (Nanotechcenter, Tambov, Russia, length 1–10 µm, diameter 8–30 nm, external specific surface—about 270 m^2^/g, 0.1–0.6 mmol/g CONH_2_—groups for amidated MWCNT (MWCNT-CONH_2_), and 0.1–1.0 mmol/g COOH—groups for carboxylated MWCNT (MWCNT-COOH)), and graphene oxide (Nanotechcenter, Tambov, Russia) were used as nanomaterials.

Neutral red (NR) (Dia-m, Russia), 2.6-dichlorophenolindophenol (DCPIP) (Dia-m, Russia), and thionine (TN) (Dia-m, Russia) were used as electron transport mediators.

Potassium nitrate (KNO_3_) (Dia-m, Russia) and sodium tetraborate pH 9.0 (NaB_4_O_7_) (Dia-m, Russia) were used for electrochemical polymerization.

For biosensor measurements, we used a sodium potassium phosphate buffer solution with pH 6.8 (33 mM KH_2_PO_4_ + 33 mM Na_2_HPO_4_, Dia-m, Russia) and a glucose–glutamate mixture (GGS).

Potassium bromine (KBr) was used to form the tablet for IR spectroscopy (OOO Dia-m, Russia).

### 2.2. Cultivation of Microorganism Cells

The yeast *Blastobotrys adeninivorans* VKM Y-2677 was provided by the All-Russian Collection of Microorganisms of the Institute of Biochemical Physics of the Russian Academy of Sciences. ES-20/60 shaker-incubator (BioSan, Riga, Latvia), MiniSpin plus (Eppendorf, Moscow, Russia), and TG16WS (Polykom LTD, Moscow, Russia) centrifuges are used for microorganism cultivation. The biomass in microtubes between measurements was stored at –25 °C.

For the cultivation of the yeast *B. adeninivorans*, the medium was used, which in its composition contained glucose—1%, peptone—0.5%, and yeast extract—0.05% of the volume of the medium. The cultivation time was 20 h (temperature 28 °C). Centrifugation was carried out at 7000× *g* for 10 min. A phosphate buffer solution (pH 6.8, 30 mM) was used to wash the biomass.

### 2.3. Thionine Electropolymerization

The working electrode was put in a 0.1 M KNO_3_ solution and electrochemically treated by cycling from 0.0 to +1.0 V potential relative to the reference electrode (Ag/AgCl) for 10 cycles until stable cyclic voltammograms were obtained. A solution with 0.025 M NaB_4_O_7_ (pH = 9.0), 1 mM thionine, and 0.1 M KNO_3_ was used for electrochemical polymerization, which was carried out by cycling from −1.0 to +1.0 V at a scan rate of 50 mV/s for 30 cycles. The conductive polymer structure is given in Figure 1.

### 2.4. Formation of the Working Electrode

The working electrode was formed by filling a plastic tube with a working surface area of 6.3 mm^2^ with the prepared graphite powder-mineral oil paste in a ratio of 100 mg of powder to 0.04 mL of oil. The electrode with poly(thionine) obtained by electrochemical polymerization was dried at room temperature for 10 min. To obtain electrodes with nanomaterials, graphite-paste electrodes were modified before electrochemical polymerization with 0.005 mL of a suspension of SWCNT, MWCNT, or graphene oxide.

A suspension of microorganisms with a volume of 0.005 mL and a content of 200 mg ww/mL was applied and dried at room temperature for 10 min to immobilize yeast cells on the surface of the modified electrode. To hold the cells on the electrode surface, a dialysis membrane was used, which was fixed with a plastic ring.

### 2.5. Registration of Current-Voltage Dependences

Cyclic voltammograms were recorded using an Ecotest-VA analyzer (Ekoniks-Expert, Moscow, Russia) in a three-electrode system. The working electrodes were graphite-paste electrodes with various modifications; a reference electrode was a saturated silver chloride electrode (Ag/AgCl); the auxiliary electrode was a platinum electrode. Here, 0.15 M phosphate buffer (pH 6.8) and 0.001 M soluble mediators (NR, DCPIP, and TN) were used for measurement. Cyclic voltammograms were obtained at a scan rate of 10 mV/s to 200 mV/s at a temperature of 22 °C. The cell volume was 15 mL.

### 2.6. Biosensor Measurements

An IPC-micro galvanopotentiostat (Volta, Novocherkassk, Russia) with two electrodes (a graphite-paste working electrode and a saturated silver chloride reference electrode) was used for recording biosensor responses. During the measurement, the electrodes were immersed in a cell with a potassium-sodium-phosphate buffer solution with a pH of 6.8.

The operating potential of the system was set as the potential of the used electron transport mediator poly(thionine) (−25 mV). The measurement temperature was 19–20 °C and the cell volume was 5 mL. After a stable current level was established, the amount of glucose-glutamate mixture necessary to obtain a given concentration was introduced into the cell with a micropipette. After each measurement, the cell was washed with a buffer solution.

### 2.7. Impedance Spectroscopy

Electrochemical measurements were carried out using a three-electrode system. The working electrodes were graphite paste electrodes of various modifications; the reference electrode was a saturated silver chloride electrode (Ag/AgCl); and the auxiliary electrode was a platinum electrode. The measurements were carried out in a cell with a volume of 15 mL at a temperature of 22 °C. A CS310M potentiostat (Corrtest, Wuhan, China) was used to record impedance. Here, 33 mM potassium phosphate buffer (pH 6.8) was used as the background solution. The operating potential of the system was set as the potential of the used electron transport mediator poly(thionine) (−25 mV), the frequency range from 40 kHz to 0.2 Hz, and 10 mV of voltage modulation amplitude. The impedance characteristics were studied. The ZView program (China) was used for selecting a suitable equivalent electrical circuit for the studied system.

### 2.8. IR Spectroscopy

IR spectra were obtained with the FMS 1201 infrared Fourier spectrometer (Monitoring, Moscow, Russia) in a KBr (Dia-m, Russia) in the 4000–500 cm^−1^ range and with a mass ratio of the conductive matrix to KBr of 2:300 (mg).

### 2.9. Scanning Electron Microscopy (SEM)

For optimization of analytical measurements, the approach was used at measurements [24]. Before analysis, the samples were added to an aluminum rod with a diameter of 25 mm, and with graphite adhesive tape, they were fixed. Magnetron sputtering was used for metal coating [25]. A thin gold/palladium (60/40) film of 10 nm was obtained. The SEM images were obtained with the emission scanning electron microscope Hitachi SU8000 (FE-SEM) at the secondary electron mode, 5 kV, and 8–10 mm working distance. The morphology of the samples was studied, taking into account the possible effect of the metal coating on the surface [25].

### 2.10. Long-Term Stability of Working Electrodes

The long-term stability of the working electrodes (days) was assessed by the daily study of the sensor responses after adding the mixture of glucose and glutamic acid (GGA) model solution. Between measurements, the electrode was stored in a buffer solution at 4 °C. During the operation of the sensor, the number of days during which the response of the sensor was at least 50% of the response obtained after the formation of the working electrode was taken.

### 2.11. Operational Stability of Working Electrodes

The operational stability was evaluated by the relative standard deviation during repeated measurements of the GGA model solution and by the residual activity (% of the original) of the electrode after several consecutive measurements over a certain period of time.

### 2.12. Determination of BOD by the Standard Dilution Method

For BOD_5_ determination, standard protocols were used [3]. The Expert-001-4.0.1 BOD thermooximeter (Econix-Expert, Russia) was used for dissolved oxygen measurement before and after sample incubation.

## 3. Results

### 3.1. Selection of Carbon Nanotubes for the Formation of Conducting Nanocomposite Systems

Graphene oxide, SWCNT, MWCNT, MWCNT–COOH, and MWCNT–CONH_2_ nanomaterials were used for graphite paste electrode modification. Graphene oxide and CNT are the most commonly used high-conductive carbon nanomaterials [26,27]. Application of CNT (single-walled and multi-walled carbon nanotubes) with different functionalizations increases the biosensor’s sensitivity [4,28,29]. DCPIP, NR, and thionine solutions are used as electron transport mediators. After electrode surface modification with nanomaterials, the working electrode surface area has been increased, and the electron transfer by mediators has been raised due to shorter diffusion distances for the redox molecules [30]. Electron transfer can be limited by electron transfer kinetics or diffusion-controlled processes. Both limiting stages are identified by cyclic voltammetry [30]. CV curves have been recorded at a scan rate of 10–100 mV/s (Figure 2), and the dependences of the limiting current of the oxidative peaks at the potential scan rate were analyzed. According to Equation (1) from the Nicholson model, the diffusion-controlled process has been realized when limiting anode current depends linearly on the root of the scan rate. If the limiting anode current increases linearly with an increase in the scan rate, then the limiting stage is electron transfer kinetic, and further calculation of heterogeneous constants was performed using the Laviron model (2) [30,31,32]. When using the Nicholson model, it is necessary to take into account the diffusion coefficient (3), which is expressed in the Randles–Shevchik Equation (4). Table 1 lists the heterogeneous electron transfer constants.
(1)$$k_{s}=\psi \sqrt{\pi \frac{nF\upsilon }{RT}D},$$
(2)$$\log\;(k_{s})=\alpha \log\;\left(1-\alpha \right)+\left(1-\alpha \right)log\alpha -\log\;\left(\frac{RT}{nF\upsilon }\right)-\frac{\alpha \left(1-\alpha \right)nF∆E}{2.3RT},$$
(3)$$D=\frac{{tg\alpha }^{2}}{{\left(2.99\times 10^{5}nAC\sqrt{an}\right)}^{2}},$$
(4)$$I_{p}=2.99\times 10^{5}nAC\sqrt{aD\upsilon n},$$
where $k_{s}$ is the heterogeneous electron transfer rate constant (s^−^^1^cm); *ψ* is a parameter connected with the peak potential difference (∆*E*, mV), calculated according to [33]; *π* is a math constant (3.14); n is the electron number; *F* is the Faraday constant (96,500 C/mol); *υ* is the scan rate (V/s); *R* is the universal gas constant (8.314 J·mol/K); *T* is temperature (K); *D* is the coefficient of diffusion (cm^2^/s); *α* is the cathode transfer coefficient; (1 − *α*) is the anode transfer coefficient; ∆*E* is the anode and cathode peak potential difference (V); *tgα*—slope of the liner curve limiting the current of the oxidative peaks on the potential scan rate; *A* is the working area of the electrode surface (cm^2^); *C* is the mediator concentration (mol/cm^3^); and *α* is the value of the transfer coefficient.

From the results, it can be concluded that the redox activity of the mediator grows in the line NR < thionine < DCPIP, the last one having the highest diffusion coefficients (Table 1). In comparison to this reported work, the diffusion coefficient values for our system have the same order as other reports [34,35], and the electrode material has an insignificant effect on the mass transport of the mediators. According to the Laviron and Nicholson model (Table 1), electron transfer limitations have been changed from diffusion-controlled processes to electron transfer kinetics after electrode modification by nanomaterials. Carbon nanomaterials have a strong adsorption capability for various organic compounds [36]. The studied electrochemical process can be controlled by adsorbed species of DCPIP, NR, and Thionine. Electron transfer kinetics by adsorbed species have been reported in the system with anthranilic acid solution and a glassy carbon electrode modified with MWCNT [37], L-tryptophan solution, and a carbon nanotube paste electrode [38].

According to heterogeneous electron transfer rate constants (Table 1), the most perspective system of carbon paste electrode is SWCNT modification and using thionine as redox-mediator particles. Low obtained data of heterogeneous electron transfer rate constant in the system with chemistry-modified MWCNT compared with non-modified MWCNT (Table 1) can be explained by changed structure after functionalization, and defect incorporation can negatively influence electron transport properties of nanomaterials [39]. The obtained data show that the heterogeneous electron transfer rate constant is higher in systems based on SWCNT, most likely due to the fact that this nanomaterial has the highest conductivity properties compared with other carbon nanomaterials [40]. The heterogeneous electron transfer rate constant of thionine and the SWCNT-modified electrode has been compared with other reported systems.

The obtained heterogeneous constants (Table 2) are lower than for nanocomposite, and the studied system has insignificant kinetic limitations compared to other reports [41,42,43,44,45,46]. For solving this problem, thionine was used as a monomer for redox-polymer formation. Redox polymers have high electrochemical conductivity properties and are applied for energy storage [47,48].

### 3.2. Formation of Nanocomposite Materials Based on Redox-Active Biocompatible Polymers

Among the polymers used to create electrochemical systems for biosensor sensors and biofuel cells, poly(thionine) can be distinguished. Due to its resistance to environmental influences, high conductivity, and biocompatibility, poly(thionine) has been successfully used in the design of biosensors for the determination of glucose and uric acid [49], clinically significant markers of lung cancer [50], NADH [51], ascorbic and uric acids [52], as well as acetaminophen [23]. To obtain a redox-active poly(thionine) polymer, thionine, which is well subjected to electropolymerization, was used as a monomer [53]. In our study, thionine showed the highest heterogeneous rate constant of electron transfer to the electrode (Table 1). The structure of the poly(thionine) obtained by electrochemical polymerization was proved using the IR spectroscopy method (Figure 3).

The peak at 2923 and 2852 cm^−1^ in the IR spectrum of pTN refers to the stretching vibrations of C-H hydrogen atoms of the aromatic ring; most often, there is no peak at 2852 cm^−1^ in the structure of the monomer, which indicates the predominant mechanism of polymer formation through the phenothiazine ring. Also, characteristic oscillation signals of C=C bonds in the ring were observed at 1610 and 1457 cm^−1^. Another signal characterizing aromatic compounds was observed at 798 cm^−1^ and denotes C-H vibrations of isolated hydrogen. The peak at 1628 cm^−1^ is associated with planar deformation -NH_2_, which also indicates the binding of the nitrogen atom of the primary amino group to the phenothiazine ring. The wide band at 3420 cm^−1^ is explained by the presence of O-H groups due to the presence of water in the sample; also in this region, there may be a peak of valence vibrations -NH_2_ groups in the region of 3100 cm^−1^. Weak absorption bands in the region of 1100–1400 cm^−1^ refer to planar deformation vibrations of C-H aromatic bonds [54]. Peaks in the region of 550–650 cm^−1^ correspond to valence fluctuations of the C-S bonds of phenothiazine. Infrared radiation proved that thionine polymerization took place and a polymer was formed on the electrode; the IR data correspond to the works [53,54].

To create a conductive nanocomposite material, the electropolymerization of thionine was carried out on the electrode surface modified by SWCNT. The structure of the studied conducting systems was studied by scanning electron microscopy. The resulting images are shown in Figure 4.

A highly developed surface of the graphite-paste electrode (Figure 4a) provides a large area for yeast contact with the conductive polymer, which should improve the sensitivity of developed biosensors. SWCNT modification (Figure 4b) increases working area and provides electron transfer. pTN modification (Figure 4c) is a 3D porous polymer film covering the electrode surface, which is invisible in Figure 4c as a result of polymer chain growth on the graphite surface [43,44].

### 3.3. Electrochemical Properties of the Created Conducting Systems

Conducting polymer pTN and its SWCNT nanocomposite base were used for yeast *Blastobotrys adeninivorans* immobilization. *Blastobotrys adeninivorans* is distinguished by its resistance to high temperatures and high osmotic pressure, as well as a large number of metabolizable substrates. Such unique features make them a promising biomaterial for receptor systems of BOD biosensors, which makes it possible to create stable biosensors showing a high correlation with the standard analysis method [54,55]. In Figure 5, different ways of yeast immobilization are presented: using only pTN polymer (Figure 5a), pTN polymer and SWCNT modification (Figure 5b), pTN polymer and NR second mediator (Figure 5c), and pTN polymer, SWCNT, and NR second mediator (Figure 5d). Two-mediator systems (Figure 5c,d) were used for more effective electron transport from yeast cells to the electrode; this approach is commonly used for yeast biosensor development. The choice of neutral red as the second mediator was based on the high rate of interaction with yeast *B. adeninivorans* [56].

For each modification of the working electrode, CV curves were recorded and are given in Figure 6.

In the process of electrode modification, the electrochemical properties change (Figure 6). The introduction of CNTs into the system sharply increases the efficiency of electron transfer, and as a result, the anodic peak increases. The introduction of cells reduces the magnitude of the anodic peak since some of the redox particles also interact with the biomaterial. When the substrate is introduced, the anodic peak increases, which indicates the efficiency of bioelectrocatalysis in the systems under study.

For the practical use of “microorganism-conducting polymer” systems, it is important to study both the electrochemical aspects of electron transfer to the electrode and the features of the interaction of the conductive polymer with the microorganism used. In the double mediator system with redox polymer, the electrocatalytic oxidation of the substrate by yeast follows the following scheme:

S + E_ox_ (*B. adeninivorans*) → P + E_red_ (*B. adeninivorans*)

E_red_(*B. adeninivorans*) + NR_ox_ → NR_red_ + E_ox_(*B. adeninivorans*) (Rate constant (k_3_) of interaction with microorganisms)

NR_red_ + pTN_ox_ → pTN_red_ + NR_ox_ (Rate constant (k_2_) of interaction redox-species)

pTN _red_(*B. adeninivorans) *→ pTN _ox_ + n ē (Rate constant of heterogeneous electron transfer (k_1_))

Using CV [57], constant rates of the electrocatalytic stages were estimated and are given in Table 3.

In order for the transfer of electrons in the two-mediator system to proceed sequentially and both mediators to not compete with each other, it is necessary to provide a difference in the rates of interaction with both the biomaterial and the electrode, so that one mediator interacts quickly with the electrode but slowly with the biomaterial, and the second, on the contrary, quickly interacts with the biomaterial but slowly with the electrode [58,59,60,61]. The efficiency of electron transfer in a two-mediator system will also depend on the rate of interaction of two mediators with each other (k_2_); a high value of this constant will not limit the process of electron transfer during the operation of biosensors under these conditions. It should be noted that the redox-active polymer poly(thionine) in the presence of SWCNTs transfers electrons to the electrode (k_1_) an order of magnitude faster than neutral red. However, neutral red reacts faster with yeast (k_3_), and neutral red with poly(thioninone) and SWCNT reacts faster (k_2_). Based on the results obtained, we can assume the sequence of electron transfers shown in Figure 5d.

The studied double mediator system based on pTN and NR applied to yeast *B. adeninivorans* by constant rates of the electrocatalytic stages (k_1_ and k_2_) is inferior to the ferrocene-NR double mediator system [62], but stabilization redox species in polymers provide a more stable long-term system.

The electrical conductivity of the working electrode is one of the most important biosensor parameters because it directly affects the electron transfer rate from the enzyme-active centers of yeast to the electrode during the substrate transformation. Electrochemical impedance spectroscopy was used to estimate the effect of SWCNTs, a redox-active polymer, microorganisms, and NR mediators on the electron transfer process. The working electrode was modified by each component separately, and for each combination of studied components, impedance spectra were obtained (Figure 7).

The equivalent electrical circuits used to fit the spectra are shown in Figure 7b,c. Pure graphite electrodes and pTN-modified electrodes, which are less complex in architecture, have been successfully described by a circuit containing Rs (electrolyte resistance) connected in series with a parallel combination of Rct and a constant phase element (CPE), where CPE is a constant phase element representing a double-layer capacitance and resistance charge transfer Rct. The constant phase element is modeled as a non-ideal capacitor and is described as CPE = 1/T-(iωC)-α, where ω is the angular frequency, α is the CPE index reflecting an inhomogeneous surface, τ is the, representing the capacitance of the CPE, and C is the capacitance. For electrodes with CNT and cells, the equivalent electrical circuit is shown in Figure 7c. This scheme contains the cell resistance Rs connected in series with a parallel combination of Rct and CPE and the Warburg finite element Zw, which indicates the presence of diffusion resistance. The Warburg element is expressed in terms of Zw = R_w_cth[(Tiω)^P^]/(Tiω)^P^, where α < 0.5 is the dimensionless index, τ is the diffusion time constant, and R_w_ is the diffusion resistance. Data obtained from the analysis of the impedance spectra are shown in Table 4.

For all electrodes, the cell resistance is 210 ± 40 Ohms. As expected, modification of a pure graphite electrode with pTN decreases the charge transfer resistance R_ct_ from 1900 kOhm to 202 kOhm, and CNTs lead to an even greater decrease in resistance up to 3 kOhm due to their high conductive properties. At the same time, when cells are added to the system, a minimum value of charge transfer resistance is achieved. It demonstrates the effectiveness of the composite as an electron carrier in a cellular biosensor.

All modifications of electrodes containing cells have similar roughness values close to 1, which indicates an increase in the heterogeneity of the surface of electrodes modified by cells compared to electrodes without cells. For all electrodes containing CNT, a higher value of C_dl_ capacitance is observed, which is probably due to an increase in the surface area of the electrodes.

It can be noted that the pTN/SWCNT + cells + NR modification has the lowest diffusion time (10 ms) and resistance Rw = 39 Ohm, which proves that it is the best electrocatalyst of the considered. When a substrate is added to this system, the diffusion time increases, which in general can be explained by the appearance of a physical barrier created by the substrate. However, the charge transfer resistance remains at the same level (0.04 Ohm). Apparently, this is a consequence of the presence of the NR solution in the system. The obtained schemes are consistent with the data obtained for polythionine and CNT-modified electrodes [63].

### 3.4. Analytical and Metrological Characteristics of BOD Biosensors Based on the Developed Conducting Systems

Calibration dependencies of the analytical signal for the BOD_5_ index were obtained (Figure 8) for biosensors based on the developed systems. As a model, we used a mixture of glucose and glutamic acid (GGA) in a mass ratio of 1:1, which is used as a standard in determining BOD_5_ in international practice [3]. In accordance with the regulatory documentation, it was assumed that a BOD_5_ equal to 205 mg/dm^3^ corresponds to a solution containing 150 mg/dm^3^ glucose and 150 mg/dm^3^ glutamic acid (BOD_5_ = 0.68 × C_GGA_).

To quantify the content of analytes in the sample, the hyperbolic dependences of the biosensor response on BOD_5_ (Figure 8) were approximated by modeling within the Michaelis–Menten kinetics (Equation (5)). Since the electrochemical response of the bioreceptor element based on whole cells is provided by the enzymatic reactions of microorganisms,
(5)$$R=\frac{R_{\max}[S]}{K_{M}{}^{}+{\left[S\right]}^{}}$$
where *R_max_* is the maximum biosensor response at [*S*]→∞, [*S*]-BOD_5_ contain; *K_M_* is the effective Michaelis constant.

According to Equation (5), at low substrate concentrations (lower than *K_M_*), the biosensor response will be linearly proportional to what BOD_5_ contains. This linear section of curves was used for BOD_5_ analysis. The lower limit of this linear part was checked statistically; the relative standard deviation was less than 0.33. The main analytical and metrological BOD biosensor parameters are given in Table 5.

According to Table 5, a biosensor based on the poly(thionine)-SWCNT-NR system has a more stable response: the relative standard deviation (operational stability) is 3.2%. With regard to long-term stability, all biosensors work on average for about 15 days, which is enough for laboratory analysis. The time of one analysis does not exceed 8 min, which allows for a significantly increased productivity of the analysis compared to the classical BOD_5_ method.

For the analysis of natural waters, it is most expedient to use the poly(thionine)-SWCNT-NR system since the lower limit of the biosensor allows estimating BOD_5_ values in the range of 0.4–62 mgO_2_/dm^3^. Thus, the bioreceptor element based on yeast *B. adeninivorans* and poly(thionine)-SWCNT-NR has the highest sensitivity and selectivity and outperforms some known BOD biosensors (Table 6).

Table 6 shows that the developed BOD biosensor based on the poly(thionine)-SWCNT-NR system is better in the range of determined concentrations. Thus, firstly, a combination based on the yeast *B. adeninivorans* and the two-mediator system poly(thionine)-SWCNT-NR can be used for more BOD-sensitive system creation that outperforms other BOD-analogues in analytical opportunity. The lower BOD limit of the developed biosensor allows for natural water sample analysis.

### 3.5. Analysis of Water Samples with a Developed Biosensor and a Standard Method for Determining BOD_5_

Eight wastewater samples were used for BOD analysis by the developed biosensor. The correlation between the BOD_5_ values determined by the standard BOD_5_ method and the values determined using the developed biosensor is given in Figure 9.

Samples with a BOD greater than 62 were tested with dilution. Statistical processing confirms the developed biosensor based on the yeast *B. adeninivorans* and the “poly(thionine)-SWCNT-NR” system can be effectively used to analyze various water samples. The BOD data of the standard 5-day method and the developed biosensor results differ insignificantly.

## 4. Conclusions

Three receptor systems based on the conductive polymer poly(thionian), SWCNT, neutral red mediator, and yeast *B. adeninivorans* were obtained in this work. Based on the analysis of certain electrochemical properties, it was found that the formed systems poly (thionine), poly(thionine)-SWCNT, and poly(thionine)-NR are sufficiently effective for further amperometric analysis. A biosensor based on the yeast *B. adeninivorans* and the poly(thionine)-SWCNT-NR system was identified as the most promising and suitable for creating a BOD biosensor. It surpasses other biosensors obtained in current work and literary analogues in the range of determined concentrations (0.4–62 mg O_2_/dm^3^). Thus, for the first time, the fundamental possibility of effectively using nanocomposite materials based on SWCNTs and the redox-active polymer poly(thionine) as one of the components of two-mediator systems for electron transfer from yeast microorganisms to the electrode has been shown, which opens up prospects for creating stable and highly sensitive electrochemical systems based on eukaryotes.

## Figures and Tables

**Figure 1 polymers-15-03335-f001:**
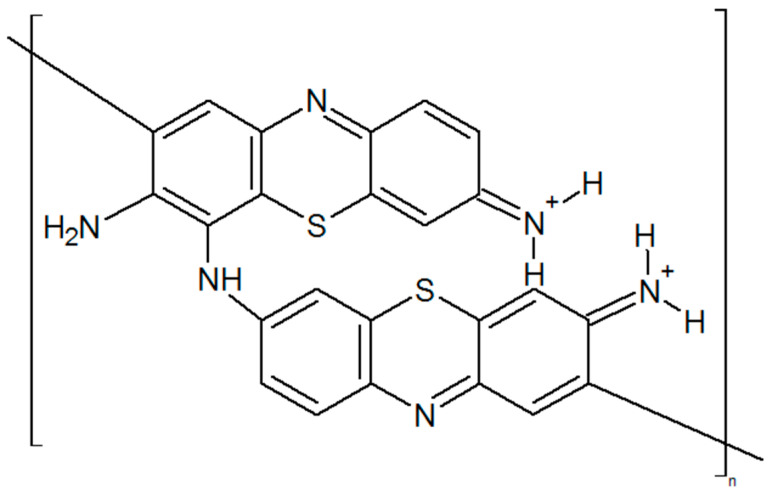
Structure of poly(thionine) obtained by electrochemical polymerization.

**Figure 2 polymers-15-03335-f002:**
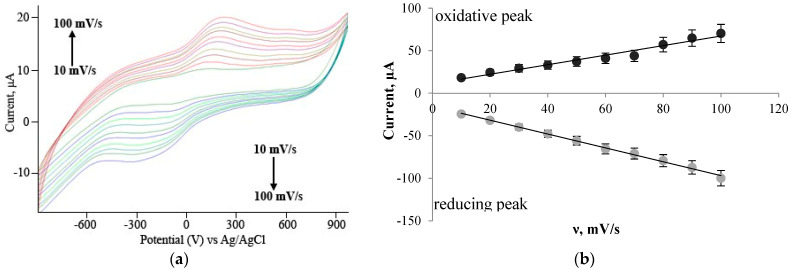
Electrochemical studies of a graphite-paste electrode modified with graphene oxide in the presence of the DCPIP mediator: (**a**) cyclic voltammogram (CV) measured at a scan rate of 10–100 mV/s in a pH = 6.8 buffer solution. (**b**) dependence for determining the limiting stage of the electron transfer process.

**Figure 3 polymers-15-03335-f003:**
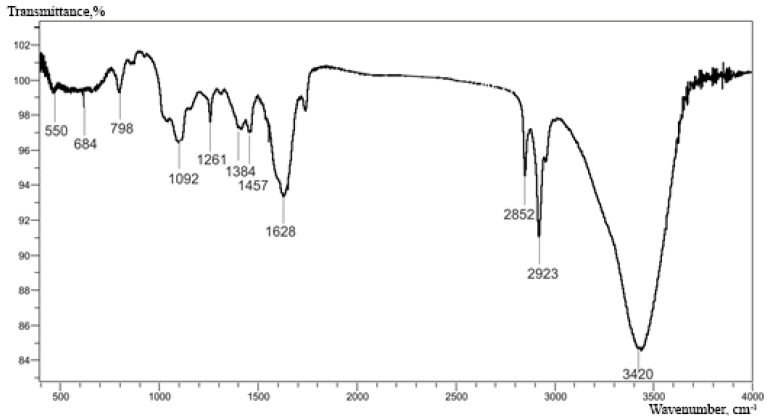
IR spectrum of the obtained pTN, taken on a tablet KBr.

**Figure 4 polymers-15-03335-f004:**
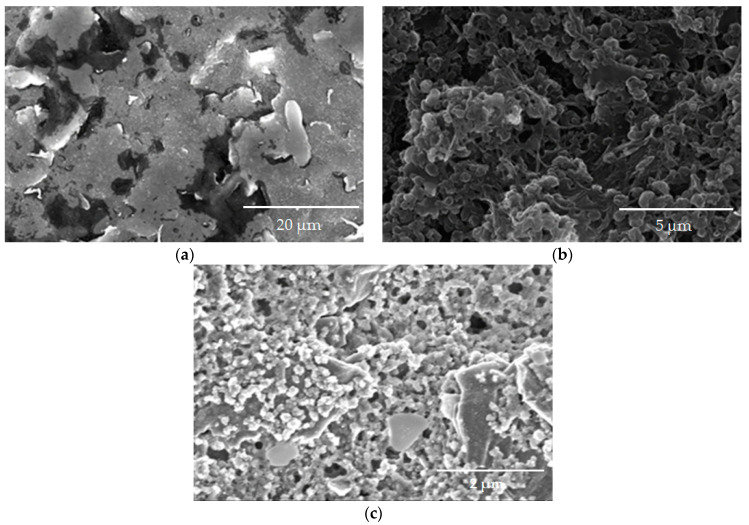
Scanning electron microscopy for electrode modification of electrode surface: (**a**) pure graphite-paste electrode; (**b**) SWCNT modification; (**c**) pTN modification.

**Figure 5 polymers-15-03335-f005:**
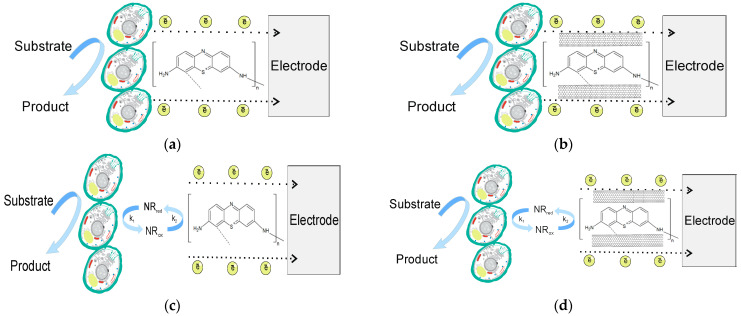
Proposed mechanism of functioning of the working electrode on the example of *B. adeninivorans* cells and systems: (**a**) pTN; (**b**) pTN-SWCNT; (**c**) pTN-NR; (**d**) pTN-SWCNT-NR.

**Figure 6 polymers-15-03335-f006:**
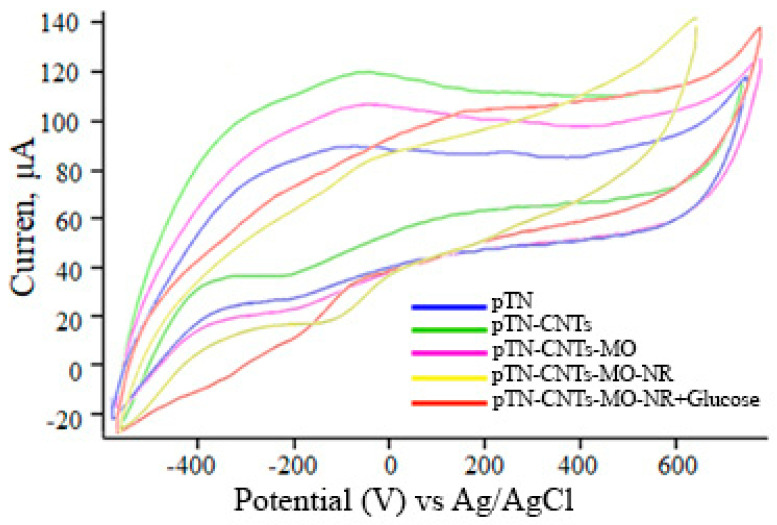
Cyclic voltammogram (CV) at different stages of electrode modification: pTN (blue line)-electrode modification with pTN polymer; pTN-CNTs (green line)-modification of the electrode with polymer pTN and CNTs; pTN-CNTs-MO (yellow line)-electrode modification with pTN polymer, CNTs, and microorganisms; pTN-CNTs-MO-NR (purple line)-modification of the electrode with polymer pTN, CNTs, microorganisms, and NR; pTN-CNTs-MO-NR + Glucose (red line)-modification of the electrode with pTN polymer, CNTs, microorganisms, and NR in the presence of a substrate. CV were measured at a scan rate of 50 mV/s in a pH = 6.8 buffer solution.

**Figure 7 polymers-15-03335-f007:**
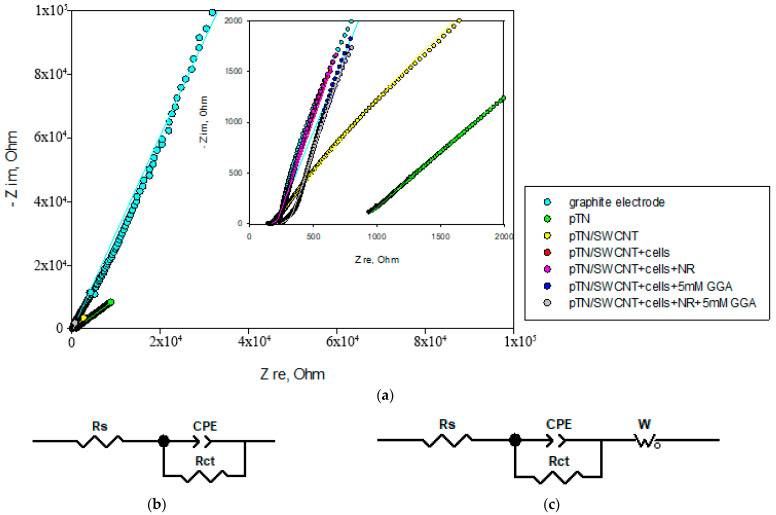
Electrochemical impedance spectroscopy. (**a**) Impedance spectra for modified electrodes, inset: approximation, at −0.25 V rel. Ag/AgCl. The *lines* show equivalent circuit fittings. (**b**,**c**) Equivalent circuit used to fit the spectra.

**Figure 8 polymers-15-03335-f008:**
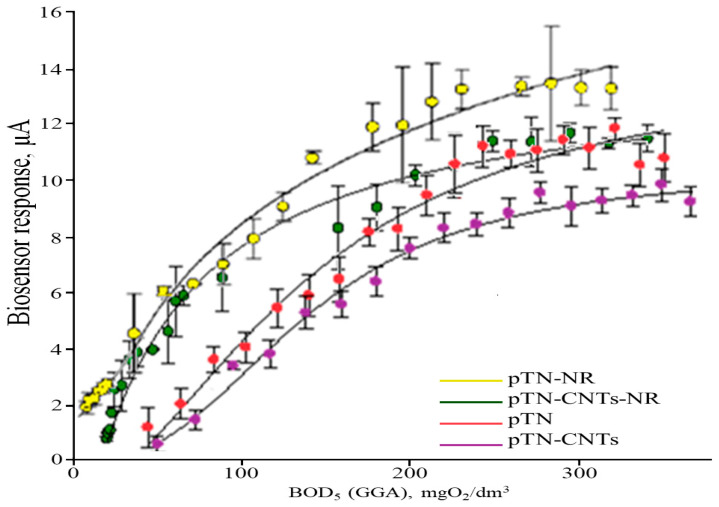
Dependence of the biosensor response on BOD5, inset: approximation, at 0.25 V redox against reference electrode Ag/AgCl.

**Figure 9 polymers-15-03335-f009:**
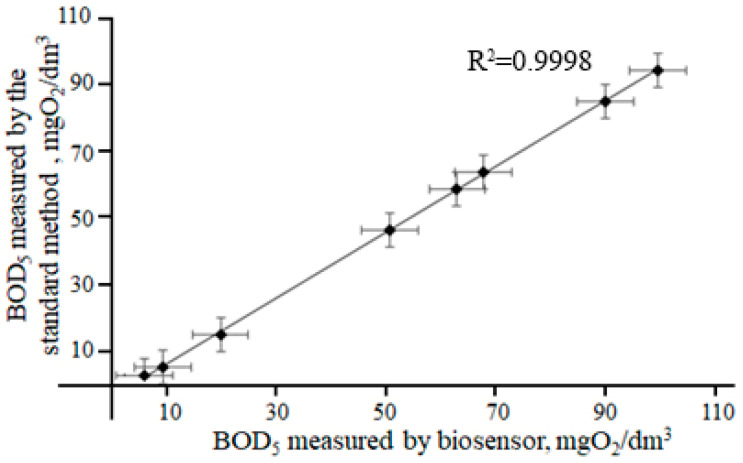
Correlation between the BOD_5_ values determined by the standard BOD_5_ method and the values determined using the developed biosensor based on the poly(thionine)-SWCNT-NR system and the yeast *B. adeninivorans*.

**Table 1 polymers-15-03335-t001:** Heterogeneous constants of electron transfer to a graphite-paste electrode modified with nanomaterials (sweep rate 100 mV/s).

Mediator	Heterogeneous Electron Transfer Rate Constant, (cm·c^−1^)	Coefficient of Diffusion (cm^2^/s)	The Rate-Limiting Step of Electron Transfer
Graphite paste electrode without modification			
DCPIP	0.0018 ± 0.003	8.57 × 10^−6^	diffusion controlled process
NR	0.000921 ± 0.000002	6.23 × 10^−6^	diffusion controlled process
Thionine	0.0026 ± 0.0004	5.21 × 10^−6^	diffusion controlled process
Graphene oxide			
DCPIP	0.081 ± 0.003	-	electron transfer kinetic
NR	0.0020 ± 0.0005	-	electron transfer kinetic
Thionine	1.012 ± 0.005	-	electron transfer kinetic
SWCNT			
DCPIP	0.39 ± 0.04	-	electron transfer kinetic
NR	0.68 ± 0.01	-	electron transfer kinetic
Thionine	1.228 ± 0.002	-	electron transfer kinetic
MWCNT			
DCPIP	0.15 ± 0.04	-	electron transfer kinetic
NR	0.062 ± 0.002	-	electron transfer kinetic
Thionine	0.31 ± 0.02	-	electron transfer kinetic
MWCNT –COOH			
DCPIP	0.089 ± 0.002	-	electron transfer kinetic
NR	0.015 ± 0.001	-	electron transfer kinetic
Thionine	0.12 ± 0.02	-	electron transfer kinetic
MWCNT –CONH_2_			
DCPIP	0.000826 ± 0.00002	-	electron transfer kinetic
NR	0.082 ± 0.005	-	electron transfer kinetic
Thionine	0.00112 ± 0.00002	-	electron transfer kinetic

**Table 2 polymers-15-03335-t002:** Heterogeneous constants of electron transfer of the most prospective system with similar reported systems.

Mediator/Nanomaterials	Heterogeneous Electron Transfer Rate Constant, (cm·c^−1^)	Reference
Thionine/SWCNT	1.228 ± 0.002	This work
Fe[(CN)_6_]^3−/4−^/MWCNT	4.38	[40]
Metanil yellow/MWCNT	11.74	[41]
NR/PbO_2_/α-Al_2_O_3_ composite	35.97 × 10^−3^	[42]
GOD/graphene-chitosan	2.83	[43]
GOD-graphene	2.68	[44]
GOD/PDDAG	1.59	[45]

**Table 3 polymers-15-03335-t003:** Kinetic characteristics of the formed systems.

Conducting System	Rate Constant of Heterogeneous Electron Transfer (k_1_), cm·s^−1^	The Rate Constant (k_2_) of the Interaction of the Conductive Polymer with the NR, dm^3^/mol × s	Rate Constant (k_3_) of Interaction with Microorganisms *B. adeninivorans*, dm^3^/(g·s)
Poly(thionine)	0.048 ± 0.005	1630 ± 80	0.00651 ± 0.00003
Poly(thionine)- SWCNT	0.130 ± 0.004	5280 ± 90	0.07089 ± 0.00002
NR [40]	0.017 ± 0.002	-	0.681 ± 0.009

**Table 4 polymers-15-03335-t004:** Data obtained from the analysis of the impedance spectra of the modified electrodes at −0.25 V rel. Ag/AgCl using the equivalent connection diagram.

Electrode	R_s_, Ohm	C_dl_, μF	α_dl_	R_ct_, kOhm	R_W_, Ohm	τ, ms	α_W_
Electrode	206	0.792	0.80	1900	-	-	-
pTN	254	76	0.53	202	-	-	-
pTN/SWCNT	182	301	0.68	3	59	5	0.37
pTN/SWCNT + cells	159	264	1	0.03	132	44	0.43
pTN/SWCNT + cells + NR	175	294	0.97	0.04	39	10	0.43
pTN/SWCNT + cells + substrate	254	259	0.98	0.06	64	16	0.43
pTN/SWCNT + cells + NR + substrate	247	214	1	0.04	118	34	0.43

**Table 5 polymers-15-03335-t005:** Characteristics of the developed biosensors.

Characteristic	pTN— *B. adeninivorans*	pTN—SWCNT— *B. adeninivorans*	pTN—NR— *B. adeninivorans*	pTN -SWCNT- NR—*B. adeninivorans*
Operational stability, %	6.1	4.7	3.9	3.2
Long-term stability, days	16	15	18	11
Duration of a single measurement, min	6–8	6–8	6–8	6–8
Linear range of determined BOD concentrations, mgO_2_/dm^3^	40–135	40–115	0.8–320	0.4–62
Correlation coefficient, R	0.9705	0.9844	0.9616	0.9729

**Table 6 polymers-15-03335-t006:** Developed biosensors and the literature analogues.

Microorganisms	System	Linear Range of Determined BOD Concentrations, mgO_2_/dm^3^	Reference
* B. adeninivorans *	Poly(thionine)-SWCNT-NR	0.4–62	This work
* E. coli *	NR	50–1000	[64]
* Activated sludge *	Hexacyanoferrate	9.8–170	[65]
* C. violaceum *	Hexacyanoferrate	20–225	[66]
* D. hanseni *	Ferrocene	25.2–320	[67]
* D. hanseni *	ferrocene–methylene blue	2.5–7.2	[68]
* S. cerevisiae *	ferricyanide–menadione	6.6–220	[69]

## Data Availability

Not applicable.

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
