# Peer review of "A Two-Mediator System Based on a Nanocomposite of Redox-Active Polymer Poly(thionine) and SWCNT as an Effective Electron Carrier for Eukaryotic Microorganisms in Biosensor Analyzers"

_polymers, 2023, doi:10.3390/polym15163335_

Round 1

Reviewer 1 Report

The manuscript by A.S.M. et al describes the development of an electrochemical system comprising single walled CNT and poly(thionine) as surface mediators for yeast ET aiming the determination of biochemical oxygen demand. The authors contribute to the development of microbial based biosensors aiming at environmental applications. The state-of-the-art is well elaborated in the introduction section and the bibliography is coherent. The scientific methodology appears to be appropriate, however, a lot of parameters are missing, and the experiments must be better explained to support the conclusions. Moreover, the English of the manuscript needs to be checked. I recommend that the manuscript be revised prior to reconsideration.

Major:

1)      The authors are required to improve the discussion over mass transport on the developed conducting nanocomposite electrodes. The way that is currently presented is confusing. Did the authors obtain the value for the diffusion coefficient for DCPIP? If they did, this needs to be included or mentioned in the manuscript.

2)      CVA method? Section 3.3 needs to be entirely reformulated because it is also confusing the way it is presented.

3)      The figures captions should include basic experimental details, such as electrolyte and redox probes present in the electrochemical cell. It is not clear.

4)      The EIS experiments need to be better explained in terms of the experimental procedure. Which was the probe employed? Why were the chosen applied potential -0.25V? The authors should consider the physical aspect of the equivalent circuit to be employed. Their construction involved a series of components which should be illustrated in the equivalent circuit to fit the spectra. The authors must correct the y-axis of Figure 7.

Minor

1)      There are a wide variety of phrases that should be corrected, a few examples: Page 2/17, line 48-49, Page 2/17, Second paragraph line 54-61, Page 3/17, line 137, and it goes on…

2)      The authors need to use one notation for dimensional units throughout the manuscript. I see mm3, uL, cm3, mL, etc… Choose one notation and stick to it.

3)      All reagents mentioned in the manuscript should be in the reagents section, there are a lot of compounds for which the abbreviation is not explained.

4)      The graphs need a better resolution.

The English of the manuscript needs to be checked. It is not acceptable in its current form. 

Reviewer 2 Report

The authors present a biosensor built with a two-mediator system of redox active polymer and CNs. The introduction is well done and is putting the reader in context with all the topics well introduced. Also the experimental part and the result and discussion are at the point: concise and well explained. I appreciate the section of the paper, mostly in the experimental, where is easy to find with a glance the information searched. Figures and conclusions are appropriate too.

I have to underline the number of self-citation of 10 out of 50 I believed what is abused, I'd reduce to 5 and think about to write a review manuscript about the topic.

Reviewer 3 Report

This paper reports on a novel biosensor that effectively measures biochemical oxygen demand (BOD). Anastasia et al. have developed a sensor capable of detecting BOD concentrations as low as 0.4 mgO2/dm^3, surpassing the performance of similar works. The authors combined conductive polymers, carbon nanotubes, and microorganisms to achieve this improvement. However, there are a few issues that need to be addressed prior to publication:

1.     Some moderate editing of the English language is required as certain parts of the content are difficult to understand.

2.     Decimal points are inconsistently represented using both dots and commas in the text. Please ensure uniformity.

3.     In Figure 2b, the labels 'cathode' and 'anode' are not the appropriate terminology.

4.     The author employs an equivalent electrical circuit model to evaluate the impact of electrode material and the interaction between cells and electrodes. However, this analysis is somewhat challenging to follow.

a.     lease consider using subscripts other than dashes. For instance, CPE-T, CPE-P, W-R, W-T, Rct, which are currently used, may not be easy to follow in the text.

b.     The analysis introduces two different models making it difficult to compare the impact of the cell. The Warburg element is a special case of constant phase element, it’s possible to merge both model into one.

c.      The analysis from line 373 to line 390 is not easily comprehensible. Please revise and make it more reader-friendly.

5.     Figure 8 is missing a legend.

6.     The calculation method for each characteristic in Table 4 is not provided, particularly for the operational stability and linear range of the determined BOD.

7.     The relationship between Figure 8 and Equation 4 is unclear. Some basic explanation is missing.

8.     In Table 4, the linear range of determined BOD for the pTN-SWCNT-NR electrode is listed as 0.4 – 62 mgO2/dm^3. However, in Figure 9, it appears that this sensor performs adequately up to around 100 mgO2/dm^3. The statement and result are not aligned.

1.     Please use the appropriate terminology. For example, "anode," "cathode," and "anodic peak" are not the correct terms to use in this context.

2.     Certain sections need revision to enhance clarity and ease of understanding. Specifically, section 3.1, the explanation of figure 6, and the interpretation of the EIS results should be revised for improved comprehension

Round 2

Reviewer 1 Report

The manuscript by A.S.M. has been revised. Unfortunately, it still presents errors and lacks a proper discussion of the obtained results to support the conclusions. Just to name a few issues: The authors were asked to improve the mass transport phenomena discussion. Not only the section became even more confusing, it still lacks data that should be at least included in the supplementary material. The authors show a table with values and explain the equations employed for the calculation of the parameters, but it still is not a proper discussion of the electrodes employed. The reference cited for the D constant is not correct.  Moreover, Figure 2 b: "inset: 241 approximation, at sweep rates of 10-100 mv/s relative to Ag/AgCl reference electrode and 242 Pt auxiliary electrode." I still don't understand what this refers to. Also, the data does not have error bars. Even though the authors changed one of the equivalent circuits used for fitting the impedance spectra, the discussion is still not properly conducted.

According to the current state of the manuscript, I cannot recommend it for publication in the Polymers journal. 

No improvements in comparison with the original version. 

Author Response

Dear reviewer!

We are grateful for your consideration of our manuscript, and we also very much appreciate your suggestions, which have been very helpful to improve it. We have reworked our article according your comments. Detailed changes are given in the table.

Reviewer’s comments

Our response

The manuscript by A.S.M. has been revised. Unfortunately, it still presents errors and lacks a proper discussion of the obtained results to support the conclusions. Just to name a few issues: The authors were asked to improve the mass transport phenomena discussion. Not only the section became even more confusing, it still lacks data that should be at least included in the supplementary material. The authors show a table with values and explain the equations employed for the calculation of the parameters, but it still is not a proper discussion of the electrodes employed.

We have fully rework section 3.1. and 3.3. Mass transport was discussed compared with other research reports; table 2 was added for extra discussion. We have remeasured CV curves for different stage of electrode formation (fig. 6).

The reference cited for the D constant is not correct.

The D constants was calculated as in the articles taking into account specific of studied system:

1) Bojang A. A., Wu H. S. Characterization of electrode performance in enzymatic biofuel cells using cyclic voltammetry and electrochemical impedance spectroscopy //Catalysts. – 2020. – V. 10. – I. 7. – P. 782.

2) Warren S. et al. Scanning electrochemical microscopy imaging of poly (3, 4-ethylendioxythiophene)/thionine electrodes for lactate detection via NADH electrocatalysis //Biosensors and Bioelectronics. – 2019. – V. 137. – С. 15-24.

The obtained parameters are compared with the report data.

Moreover, Figure 2 b: "inset: 241 approximation, at sweep rates of 10-100 mv/s relative to Ag/AgCl reference electrode and 242 Pt auxiliary electrode." I still don't understand what this refers to. Also, the data does not have error bars.

Fig. 2b was reworked, error bars were added, name of the fig. was corrected.

Even though the authors changed one of the equivalent circuits used for fitting the impedance spectra, the discussion is still not properly conducted.

Discussion of impedance parts has been broadened and compared with other research reports.

No improvements in comparison with the original version.

English errors have been corrected in the line 25, 41, 47, 52, 58, 62, 65, 326, 327, 452, 464, 473, 488, 495, 500, 510.

Round 3

Reviewer 1 Report

The manuscript has been revised and improved from previous versions. 

N.a.